# LATENT VARIABLES ON SPHERES FOR SAMPLING AND SPHERICAL INFERENCE

## ABSTRACT

Variational inference is a fundamental problem in Variational Auto-Encoder (VAE). By virtue of high-dimensional geometry, we propose a very simple algorithm completely different from existing ones to solve the inference problem in VAE. We analyze the unique characteristics of random variables on spheres in high dimensions and prove that the Wasserstein distance between two arbitrary datasets randomly drawn from a sphere are nearly identical when the dimension is sufficiently large. Based on our theory, a novel algorithm for distribution-robust sampling is devised. Moreover, we reform the latent space of VAE by constraining latent random variables on the sphere, thus freeing VAE from the approximate optimization pertaining to the variational posterior probability. The new algorithm is named as Spherical Auto-Encoder (SAE), which is in essence the vanilla autoencoder with the spherical constraint on the latent space. The associated inference is called the spherical inference, which is geometrically deterministic but is much more robust to various probabilistic priors than the variational inference in VAE for sampling. The experiments on sampling and inference validate our theoretical analysis and the superiority of SAE.

## 1 INTRODUCTION

Deep generative models, such as Variational Auto-Encoder (VAE) (Kingma & Welling, 2013; Rezende et al., 2014) and Generative Adversarial Network (GAN) (Goodfellow et al., 2014), play more and more important role in machine learning and computer vision. However, the problem of variational inference in VAE is still challenging, especially for high-dimensional data like images.

To be formal, let $\mathbb{X} = \{\boldsymbol{x}_1, \ldots, \boldsymbol{x}_n\}$ denote the set of observable data points and $\mathbb{Z} = \{\boldsymbol{z}_1, \ldots, \boldsymbol{z}_n\}$ the set of desired latent vectors, where $\boldsymbol{x}_i \in \mathbb{R}^{d_x}$ and $\boldsymbol{z}_i \in \mathbb{R}^{d_z}$. Let $p_g(\boldsymbol{x}|\boldsymbol{z})$ denote the likelihood of generated sample conditioned on latent variable $\boldsymbol{z}$ and $p(\boldsymbol{z})$ the prior, where $g$ denotes the decoder . The encoder $f$ in VAE parameterizes the variational posterior $q_f(\boldsymbol{z}|\boldsymbol{x})$ in light of the lower bound of the marginal log-likelihood

$$\log p_g(\boldsymbol{x}) = \log \int p_g(\boldsymbol{x}|\boldsymbol{z})p(\boldsymbol{z})d\boldsymbol{z} = \log \int \frac{q_f(\boldsymbol{z}|\boldsymbol{x})}{q_f(\boldsymbol{z}|\boldsymbol{x})} p_g(\boldsymbol{x}|\boldsymbol{z})p(\boldsymbol{z})d\boldsymbol{z} \tag{1}$$

$$\geq -D_{\text{KL}}[q_f(\boldsymbol{z}|\boldsymbol{x})||p(\boldsymbol{z})] + \mathbb{E}_q[\log p_g(\boldsymbol{x}|\boldsymbol{z})]. \tag{2}$$

The first term $D_{\text{KL}}[q_f(\boldsymbol{z}|\boldsymbol{x})||p(\boldsymbol{z})]$ constrains the encoded latent codes to the prior via the KL-divergence, and the second term $\mathbb{E}_q[\log p_g(\boldsymbol{x}|\boldsymbol{z})]$ serves to guarantee the reconstruction accuracy of inputs. For a Gaussian $p_g(\boldsymbol{x}|\boldsymbol{z})$ of diagonal covariance matrix, $\log p_g(\boldsymbol{x}|\boldsymbol{z})$ reduces to the variance-weighted squared error (Doersch, 2016).

The lower-bound approximation of the log-likelihood provides a feasible solution for VAE. But it also causes new problems. For example, the generated sample $g(\boldsymbol{z})$ deviates from the real distribution of $\mathbb{X}$ when sampling from the given prior due to that the learnt $q_f(\boldsymbol{z}|\boldsymbol{x})$ is incapable of matching the prior distribution well. Besides, the reconstruction $g(f(\boldsymbol{x}))$ is not satisfactory either. For imagery data, bluriness usually occurs.

In order to manipulate real images for GAN models, we usually need to formulate an encoder via the framework of VAE. The variational inference also applies in this scenario. The problems of VAE are the obstacles of putting the GAN encoder in the right way either. There are other methods of learning

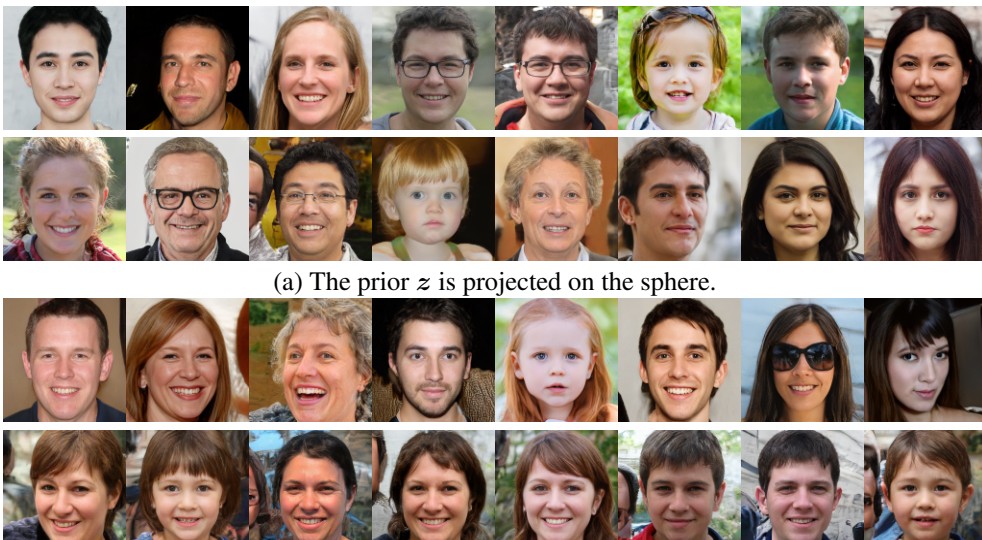

(a) The prior $z$ is projected on the sphere.

(b) The prior $z$ is not projected on the sphere.

Figure 1: Generated faces by StyleGAN with different priors. For (a) and (b), the first row shows the generated faces with the normal distribution and the second row displays the faces with the uniform distribution. The priors are both normal when training.

an encoder for GAN in the adversarial way such as (Dumoulin et al., 2017; Li et al., 2017; Ulyanov et al., 2017; Heljakka et al., 2018). However, the structural precision of reconstruction is generally inferior to the VAE framework, because the high-level semantics of objects outweigh low-level structures for such methods (Donahue & Simonyan, 2019). Besides, the concise architecture of VAE is more preferred in this scenario. Therefore, learning precise latent variables $z = f(x)$ is critical to applications of VAE and GAN.

Using a different theory in this paper, we propose a simple method to circumvent the problem. Our contributions are summarized as follows. 1) We introduce the volume concentration of high-dimensional spheres. Based on the concentration property, we point out that projecting on a sphere for data that are distributed according to the spherical mass produces little difference from the viewpoint of the volume in high-dimensional spaces. Thus, it is plausible to perform inference pertaining to VAE on the sphere. 2) We further analyze the probability distribution of distances between two arbitrary sets of random points on the sphere in high dimensions and illustrate the phenomenon of distance convergence. Furthermore, we prove that the Wasserstein distance between two arbitrary datasets randomly drawn from a high-dimensional sphere are nearly identical, meaning that the data on the sphere are distribution-robust for generative models with respect to Wasserstein distance. 3) Based on our theoretical analysis, we propose a very simple algorithm for sampling generative models. The same principle is also harnessed to reformulate VAE. The spherical normalization is simply put on latent variables instead of variational inference while preserving randomness of latent variables by centerization. In contrast to VAE and variational inference, we name such an autoencoder as Spherical Auto-Encoder (SAE) and the associated inference as spherical inference. 4) We perform extensive experiments to validate our theoretical analysis and claims with sampling and inference.

## 2    LATENT VARIABLES ON SPHERE

For latent variables or data points sampled from some priors, the projection on the unit sphere can can be easily performed by

$$z \leftarrow z/\|z\|. \tag{3}$$

This spherical normalization for priors fed into the generator is employed in StyleGAN that is the phenomenal algorithm in GANs (Karras et al., 2018b). To test the robustness of StyleGAN against the diverse distributions, we conduct two groups of experiments with the input $z$ sphere-normalized and not sphere-normalized when training StyleGAN. As shown in Figure 1, the diversity of generated

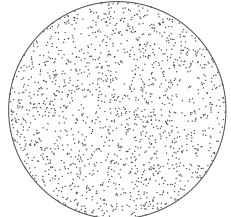 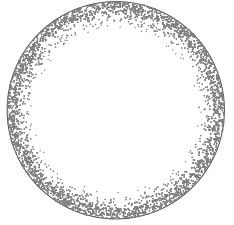

(a) Low-dimensional space.     (b) High-dimensional space.

Figure 2: Volume of spheres in different dimensional spaces. The volume of the sphere in the high-dimensional space is highly concentrated near the surface. The interior is nearly empty.

faces is good for two different distributions with normalized $\boldsymbol{z}$, whereas the face modes become similar for the case of the uniform distribution that $\boldsymbol{z}$ is not normalized. This experiment indicates that StyleGAN with sphere-normalized $\boldsymbol{z}$ is much more robust to the variation of variable modes from different distributions.

Inspired by this comparison, we interpret the benefit of using random variables on spheres by virtue of high-dimensional geometry in this section. Based on these theories, a novel algorithm is proposed for random sampling and spherical inference for GAN and VAE.

## 2.1 VOLUME CONCENTRATION

For high-dimensional spaces, there are many counter-intuitive phenomena that will not happen in low-dimensional spaces. For a convenient analysis, we assume that the center of the sphere $\mathbb{S}^d$ embedded in $\mathbb{R}^{d+1}$ is at the origin. We first present the concentration property of sphere volume in $\mathbb{R}^{d+1}$. One can find the proof in (Blum et al., 2020).

**Theorem 1.** *Let $V(r)$ and $V\left((1-\epsilon)r\right)$ denote the volumes of the two concentric spheres of radius $r$ and $(1-\epsilon)r$, respectively, where $0 < \epsilon < 1$. Then*

$$V\left((1-\epsilon)r\right)/V(r) = (1-\epsilon)^d. \tag{4}$$

*And if $\epsilon = t/d$, $V\left((1-\epsilon)r\right)/V(r) \to e^{-t}$ when $d \to \infty$, where $t$ is a constant.*

Theorem 1 says that the volume of the $d$-dimensional sphere of radius $(1-\epsilon)r$ rapidly goes to zero when $d$ goes large, meaning that the interior of the high-dimensional sphere is empty. In other words, nearly all the volume of the sphere is contained in the thin annulus of width $\epsilon r = rt/d$. The width becomes very thin when $d$ grows. For example, the annulus of width that is 0.9% of the radius contains 99% of all the volume for the sphere in $\mathbb{R}^{512}$. To help understand this counter-intuitive geometric property, we make a schematic illustration in Figure 2.

The probabilistic operations can be very beneficial from the volume concentration of spheres. Suppose that we perform the probabilistic optimization pertaining to latent variables sampled according to the distribution of the sphere volume. The probability mass is also empty due to the volume concentration. Therefore, the error is controllable if we perform it *on* the sphere, provided that these latent variables lie in high-dimensional spaces. For VAE, therefore, we can write a feasible approximation

$$\log p_g(\boldsymbol{x}) = \log \int_{\text{Int}(\mathbb{S}^d)} p_g(\boldsymbol{x}|\boldsymbol{z})p(\boldsymbol{z})d\boldsymbol{z} \approx \log \int_{\mathbb{S}^d} p_g(\boldsymbol{x}|\boldsymbol{z})p(\boldsymbol{z})d\boldsymbol{z}, \tag{5}$$

where $\text{Int}(\mathbb{S}^d)$ denotes the interior of $\mathbb{S}^d$, $\|\boldsymbol{z}\| \leq r$, and $d$ is sufficiently large. The spherical approximation for $\log p_g(\boldsymbol{x})$ is an alternative scheme except the lower bound approximation presented in equation (1). In fact, the distributions defined on the sphere have been already exploited to re-formulate VAE, such as the von Mises-Fisher distribution (Davidson et al., 2018; Xu & Durrett, 2018). But the algorithms proposed in (Davidson et al., 2018; Xu & Durrett, 2018) still fall into the category using the variational inference like the vanilla VAE. To eliminate this constraint, we need more geometric analysis presented in the following section.

## 2.2 DISTANCE CONVERGENCE

To dig deeper, we examine the pairwise distance between two arbitrary points randomly sampled on $\mathbb{S}^d$. The following important lemma was proved by (Lord, 1954; Lehnen & Wesenberg, 2002).

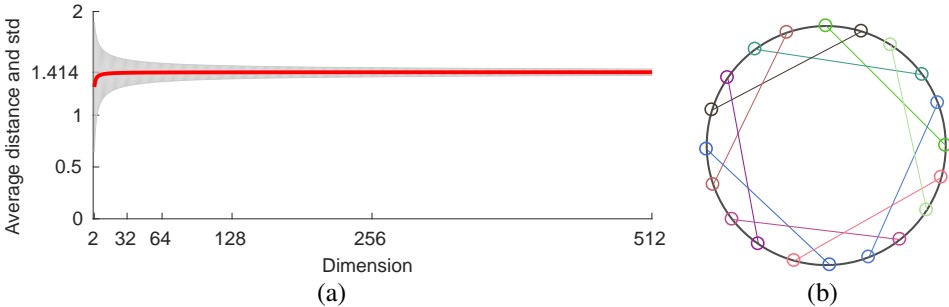

Figure 3: The illustration of average distance between two points randomly sampled on unit spheres of various dimensions. (a) The average distance (red curve) and standard deviation (gray background). (b) Schematic illustration of pairwise distances between two arbitrary points in high dimension. They are nearly identical.

**Lemma 1.** *Let $\xi$ denote the Euclidean distance between two points randomly sampled on the sphere $\mathbb{S}^d$ of radius $r$. Then the probability distribution of $\xi$ is*

$$\rho(\xi) = \frac{\xi^{d-2}}{c(d)r^{d-1}}\left[1-\left(\frac{\xi}{2r}\right)^2\right]^{\frac{d-3}{2}}, \tag{6}$$

*where the coefficient $c(d)$ is given by $c(d) = \sqrt{\pi}\Gamma\left(\frac{d-1}{2}\right)/\Gamma\left(\frac{d}{2}\right)$. And the mean distance $\xi_\mu$ and the standard deviation $\xi_\sigma$ are*

$$\xi_\mu = \frac{2^{d-1}r\left[\Gamma\left(\frac{d}{2}\right)\right]^2}{\sqrt{\pi}\Gamma\left(d-\frac{1}{2}\right)} \quad and \quad \xi_\sigma = \sqrt{2}r\sqrt{1-\frac{\xi_\mu^2}{2r^2}}, \tag{7}$$

*respectively, where $\Gamma$ is the Gamma function. Furthermore, $\xi_\mu \to \sqrt{2}r\left(1-\frac{1}{8d}\right)$ and $\xi_\sigma \to \frac{r}{\sqrt{2d}}$ when $d$ goes large.*

Lemma 1 tells that the pairwise distances between two arbitrary points randomly sampled on $\mathbb{S}^d$ approach to be mutually identical and converge to the mean $\xi_\mu = \sqrt{2}r$ when $d$ grows. The associated standard deviation $\xi_\sigma \to 0$. This result is in some extent surprising compared to the intuition in low-dimensional spaces. We display the average distance and its standard deviation in Figure 3, showing that the convergence process is fast. Taking $\mathbb{R}^{512}$ for example, we calculate that $\xi_\mu = 1.4139$ and $\xi_\sigma = 0.0313$. The standard deviation is only $2.21\%$ of the average distance, meaning that the distance discrepancy between arbitrary $z_i$ and $z_j$ is rather small. This surprising phenomenon is also observed for neighborly polytopes when solving the sparse solution of underdetermined linear equation (Donoho, 2005) and for nearest neighbor search in high dimensions (Beyer et al., 1999).

With Lemma 1, we can study the property of two different random datasets on $\mathbb{S}^d$, which serves to distribution-free sampling and spherical inference in generative models. To this end, we first introduce the *computational* definition of Wasserstein distance. Let $\mathbb{Z} = \{z_1, \ldots, z_n\}$ and $\mathbb{Z}' = \{z'_1, \ldots, z'_n\}$ be the datasets of random variables drawn from $\mathbb{S}^d$ at *random*, respectively. Then the 2-Wasserstein distance is defined as

$$W_2^2(\mathbb{Z}, \mathbb{Z}') = \min_{\boldsymbol{\omega}} \sum_{i=1}^n \sum_{j=1}^n \omega_{ij}\|z_i - z'_j\|^2 \tag{8}$$

$$\text{s.t.} \quad \sum_{i=1}^n \omega_{ij} = \sum_{j=1}^n \omega_{ij} = 1, \tag{9}$$

where $\boldsymbol{\omega}$ is the doubly stochastic matrix. By Lemma 1, it is not hard to derive the following Theorem.

**Theorem 2.** [1] $W_2(\mathbb{Z}, \mathbb{Z}') \to \sqrt{2n}r$ *with zero standard deviation when* $d \to \infty$.

Theorem 2 says that despite the diverse distributions, the 2-Wasserstein distance between two arbitrary sets of random variables on the sphere converges to a constant when the dimension is sufficiently large. For generative models, this unique characteristic for datasets randomly sampled from high-dimensional spheres bring great convenience for distribution-robust sampling and spherical inference. For example, if $\mathbb{Z}$ and $\mathbb{Z}'$ obeys the different distributions, the functional role of $\mathbb{Z}'$ nearly coincides

---

[1]It suffices to note that the case described in Theorem 2 is essentially different from the approximate solution of Wasserstein distance via Monte Carlo sampling, where the points sampled from the unit sphere are applied as projection subspaces.

with that of $\mathbb{Z}$ with respect to Wasserstein distance, provided that both $\mathbb{Z}$ and $\mathbb{Z}'$ are randomly drawn from the high-dimensional sphere. The specific distributions of $\mathbb{Z}$ and $\mathbb{Z}'$ affect the result negligibly under such a condition. We will present the specific application of Theorem 2 in the following section.

In fact, we can obtain the bounds of $W_2(\mathbb{Z}, \mathbb{Z}')$ using the proven proposition about the nearly-orthogonal property of two random points on high-dimensional spheres (Cai et al., 2013). However, Theorem 2 is sufficient to solve the problem raised in this paper. So, we bypass this analysis to simplify the theory for easy readability.

## 3 Algorithm for sampling and inference

We will detail the distribution-robust algorithms for sampling and spherical inference. A new simplified form of VAE will be presented in this section as well.

### 3.1 Sampling

To acquire generative results from VAEs and GANs, we need to sample random vectors from a pre-defined prior and then feed them into the decoder or the generator. According to Theorem 1 and Theorem 2, however, this prior limitation can be eliminated if we project these random vectors on the corresponding sphere. To achieve this, we perform two-step manipulations on the sampled dataset from arbitrary prior distributions. The procedure is detailed in Algorithm 1. The centerization operation is motivated from the central limit theorem in probability, which transforms the distribution-specific $\mathbb{Z}$ to be nearly distribution-agnostic (on the sphere). The spherization is to project these centerized vectors on the unit sphere. We find that in practice, this simple algorithm works well to reduce the bias caused by various distributions or data modes for VAEs and GANs.

---

**Algorithm 1** Distribution-robust sampling for generative models

---

1: Sample $\mathbb{Z} = \{z_1, \ldots, z_n\} \sim P(z)$  $\qquad\qquad\qquad\qquad$ ▷ $P(z)$ is an arbitrary distribution
2: Centerization by $z_i^j \leftarrow z_i^j - \frac{1}{d}\sum_j z_i^j$ for each $z_i$ $\qquad\qquad$ ▷ $z_i^j$ is the $j$-th entry of $z_i$
3: Spherization by $\tilde{z}_i \leftarrow z_i/\|z_i\|$ for each $z_i$
4: Return $\tilde{\mathbb{Z}} = \{\tilde{z}_1, \ldots, \tilde{z}_n\}$

---

### 3.2 Spherical inference

According to Theorem 2, we may know that sampling is robust to random variables if they are randomly sampled from the high-dimensional sphere. Theorem 1 guarantees that the error can be negligible even if they deviate from the sphere, as long as they are distributed near the spherical surface. This tolerance to various modes of random variables allow us to devise a simple solution to replace the variational inference for VAE, i.e. the spherical inference we call. To be specific, we only need to constrain the centerized latent variables on the sphere, as opposed to the conventional way of employing the KL-divergence $D_{\mathrm{KL}}[q_f(z|x)||p(z)]$ and its variants with diverse priors. The sequential mappings of the autoencoder under our framework can be shown by

$$\underbrace{x \xrightarrow{f} z}_{\text{encoder}} \longmapsto \underbrace{(z - \hat{z}\mathbf{1}) \longmapsto z_i/\|z_i\|}_{\text{spherical constraint on the latent space}} \longmapsto \underbrace{\tilde{z} \xrightarrow{g} \tilde{x}}_{\text{decoder}}, \tag{10}$$

where $\hat{z} = \frac{1}{d}\sum_j z_i^j$ and $\mathbf{1}$ is the all-one vector. We can write the objective function for this type of autoencoder as

$$\min_{f,g} \|x - \tilde{x}\|_{\ell_p}^2, \ \text{ s.t. spherical constraint on } z, \tag{11}$$

where $\ell_p$ denotes the $p$-norm. The objective function of our algorithm is much simpler than that of VAE and its variants based on the variational inference or various sophisticated regularizers on the latent space.

It is clear that we utilize the geometric characteristics of latent variables on the sphere rather than some additional losses to optimize the latent space. Our algorithm is geometric and free from the probability optimization whose performance is usually limited with the approximation dilemma. In fact, the framework of VAE in (10) reduces to a standard autoencoder with the spherical constraint. There is no variational inference needed here. To highlight this critical difference, we call our algorithm Spherical Auto-Encoder (SAE).

## 4 RELATED WORK

Little attention has been paid on examining geometry of latent spaces in the field of generative models. So we find few works directly related to ours. Most relevant ones are the application of von Mises-Fisher (vMF) distribution as the probability prior (Davidson et al., 2018; Xu & Durrett, 2018). The vMF distribution is defined on the sphere. The sampling and variational inference are both performed with latent variables drawn on the sphere. However, the algorithms proposed in (Davidson et al., 2018; Xu & Durrett, 2018) both rely on the variational inference as VAE does with inequality (1). For our algorithm, the whole framework is deterministic and there is no approximation involved for inferring latent codes.

For sampling, our geometric analysis is directly inspired by ProGAN Karras et al. (2018a) and StyleGAN (Karras et al., 2018b) that have already applied the spherical normalization for sampled inputs. We study the related theory and extend the case to arbitrary distributions for both GANs and VAEs. Another related method is to sample priors along the great circle when performing the interpolation in the latent space for GANs (White, 2016). The empirical results show that such sampling yields more smooth interpolated generation. This algorithm is perfectly compatible with our theory and algorithm. Therefore, it can also be harnessed in our algorithm when performing interpolation as well.

Wasserstein Auto-Encoder (WAE) (Tolstikhin et al., 2018) is an alternative way of optimizing the model distribution and the prior distribution using Wasserstein distance. SAE is different from WAE because we do not really use Wasserstein distance for computation in the latent space. We just leverage Wasserstein distance to establish Theorem 2 for the theoretical analysis. Adversarial Auto-Encoder (AAE) (Makhzani et al., 2015) is another interesting method of replacing the variational inference with adversarial learning in the latent space. But both WAE and AAE need some priors to match, which are essentially different from SAE. $\beta$-VAE (Higgins et al., 2017) improves the flexibility of VAE by using a regularization coefficient to modulate the capacity of latent information. However, $\beta$-VAE is restricted by priors like VAE.

## 5 EXPERIMENT

We conduct the experiments to test our theory and algorithms in this section. Three aspects pertaining to generative algorithms are taken into account, including sampling GANs, learning the variants of autoencoder, and sampling the decoders.

The FFHQ dataset (Karras et al., 2018b) is a more complex face dataset with large variations of faces captured in the wild. We test VAE and our SAE algorithm with this benchmark dataset. We use the image size of $128 \times 128$, which is larger than the commonly chosen size in the related work and also more challenging than $64 \times 64$ or $32 \times 32$ for (variational) autoencoders to reconstruct.

### 5.1 SAMPLING GAN

Our first experiment is to test the sampling effect using four distributions. We employ StyleGAN trained with random variables sampled from the normal distribution. The other three distributions are opted to test the generation with different priors for training, i.e. the uniform, Poisson, and Chi-squared distributions. The shapes of these three distributions are significantly distinctive from that of the normal distribution. Thus, the generalization capability of the generative model can be effectively unveiled when fed with priors that are not involved during training. We follow the experimental protocol in (Karras et al., 2018a;b) that StyleGAN is trained on the FFHQ face dataset and Fréchet inception distance (FID) (Borji, 2018) is used as the quality metrics of generative results. We take $d_z = 512$, which is set in StyleGAN. This dimension is also used for both VAE and SAE for face data.

From Table 1, we can see that the generative results by the normal distribution is significantly better than the others when tested with the original samples. The uniform distribution is as good as the normal distribution when projected on the sphere. This is because the values for each random vector are overall symmetrically distributed according to the origin. They satisfy the condition in Theorem 2 after the spherical projection. The accuracy of Poisson and Chi-squared distributions is considerably improved after centerization, even better than the vanilla uniform distribution. But the

Table 1: Comparison of sampling. The quantitative results are FIDs.

| | vanilla sampling | | centerization after sampling | |
|---|---|---|---|---|
| Distribution | no spherization | spherization | no spherization | spherization |
| Normal | 6.20 | 6.16 | 6.27 | **6.12** |
| Uniform | 33.93 | 6.16 | 33.86 | **6.16** |
| Poisson | 23.70 | 26.85 | 18.15 | **6.19** |
| Chi-squared | 25.26 | 27.07 | 11.34 | **6.16** |

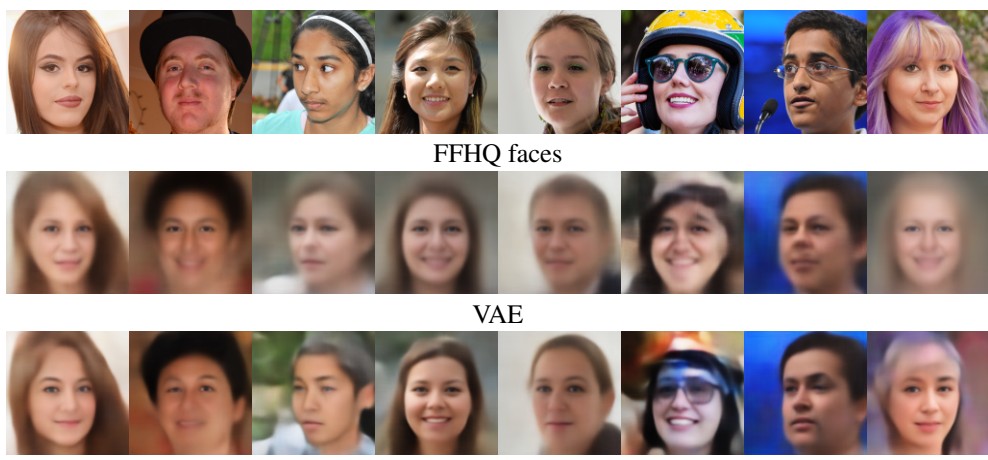

FFHQ faces

VAE

SAE (ours)

Figure 4: Reconstructed faces by VAE and SAE. SAE only uses the spherical constraint in equation (10) instead of the variational inference in VAE.

accuracy difference between all the compared distributions is rather negligible after centerization and spherization, empirically verifying the theory presented in Theorem 2.

## 5.2 AUTOENCODERS

We compare the vanilla VAE with the normal distribution (Kingma & Welling, 2013) with our SAE algorithm for reconstruction and sampling tasks[2].

From Figure 4, we can see that the face quality of SAE outperforms that of VAE. The imagery details like semantic structures are preserved much better for SAE. For example, the sunglasses in the sixth image is successfully recovered by SAE, whereas VAE distorts the face due to this occlusion. It is worth emphasizing that the blurriness for images reconstructed by SAE is much less than that by VAE, implying that the spherical inference is superior to the variational inference in VAE. The different accuracy measurements in Table 2 also indicate the consistently better performance of SAE.

To test the generative capability of the models, we also perform the experiment of sampling the decoders as done in section 5.1. Prior samples are drawn from the normal, uniform, Poisson, and Chi-squared distributions, respectively, and then fed into the decoders to generate faces. Figure 5 illustrates the generated faces of significantly different quality with respect to four types of samplings. The style of the generated faces by SAE keeps consistent, meaning that SAE is rather robust to different probability priors. This also empirically verifies the correctness of Theorem 2 by solving the real problem. As a comparison, the quality of the generated faces by VAE varies with probability priors. In other words, VAE is sensitive to the outputs of the encoder with the variational inference, which is probably the underlying reason of the difficulty of training VAE with sophisticated architectures. We also present the experimental results on MNIST and CelebA in Appendix.

---

[2]We fail to train a convergent model for the spherical VAE (S-VAE) with von Mises-Fisher distribution on the FFHQ dataset. So this algorithm is not compared here. The experiment on MNIST is provided in Appendix.

Table 2: Quantitative comparison of face reconstruction.

| Metric | FID | SWD | MSE |
|---|---|---|---|
| VAE | 134.22 | 77.68 | 0.091 |
| SAE (ours) | 91.02 | 56.58 | 0.063 |

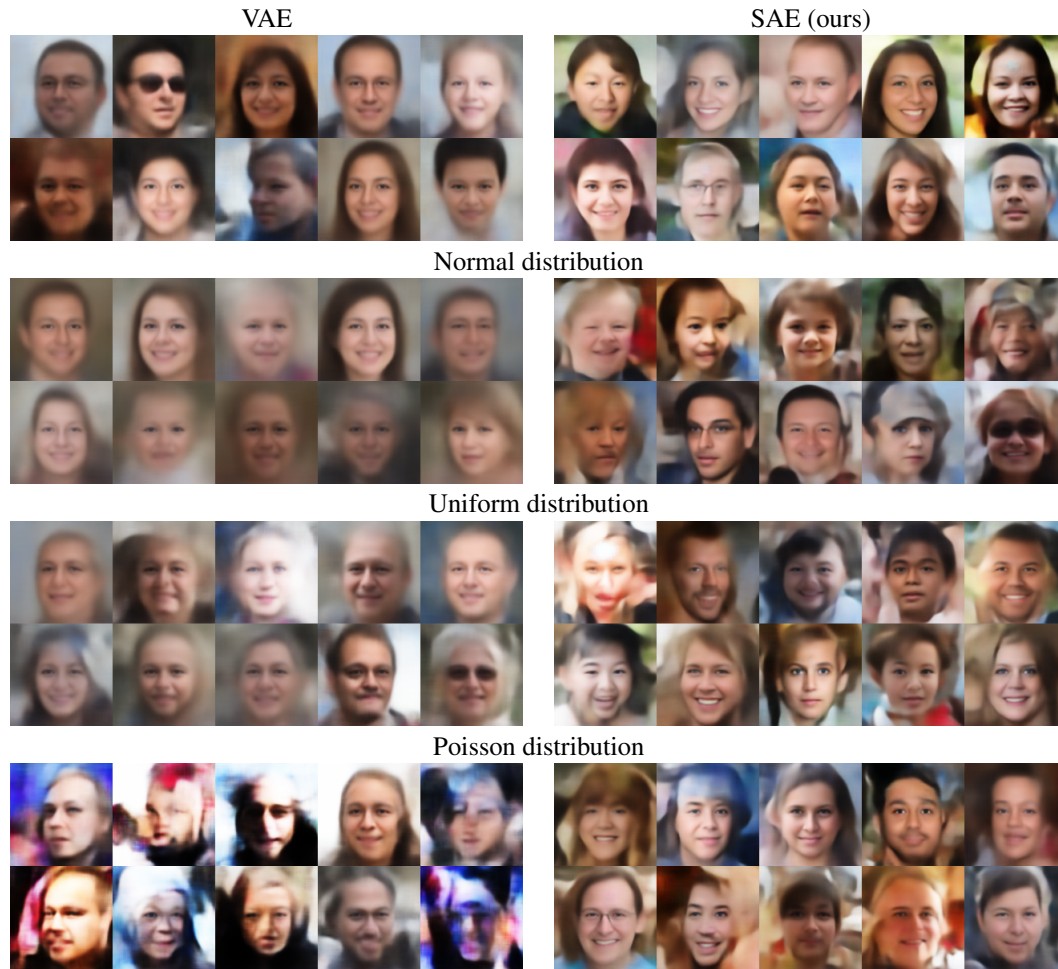

Figure 5: Generated faces with inputs of different priors. With the pre-trained decoders of VAE and SAE, the faces are generated with the random vectors sampled from the four probability priors.

## 6 CONCLUSION

In this paper, we attempt to address the issue of the variational inference in VAE and the limitation of prior-sensitive sampling in GAN. By analyzing the geometry of volume concentration and distance convergence on the high-dimensional sphere, we prove that the Wasserstein distance converges to be a constant for two datasets randomly sampled from the sphere when the dimension goes large. Based on this unique characteristic, we propose a very simple algorithm for sampling and spherical inference. The sampled data from priors are first centerized and then projected onto the unit sphere before being fed into decoders (or generators). Such random variables on the sphere are robust to the diverse prior distributions. With our theory, the vanilla VAE can be reduce to a standard autoencoder with the spherical constraint on the latent space. In other words, the conventional variational inference in VAE is replaced by the simple operations of centerization and spherization. The new autoencoder is named as Spherical Auto-Encoder (SAE). The experiments on the FFHQ face data validate the effectiveness of our new algorithm for sampling and spherical inference. It is worth noting that the applications of our theory and the novel algorithm are not limited for VAEs and GANs. Interested readers may explore the possibility in their scenarios.

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

# A APPENDIX

## A.1 RECONSTRUCTION ON FFHQ

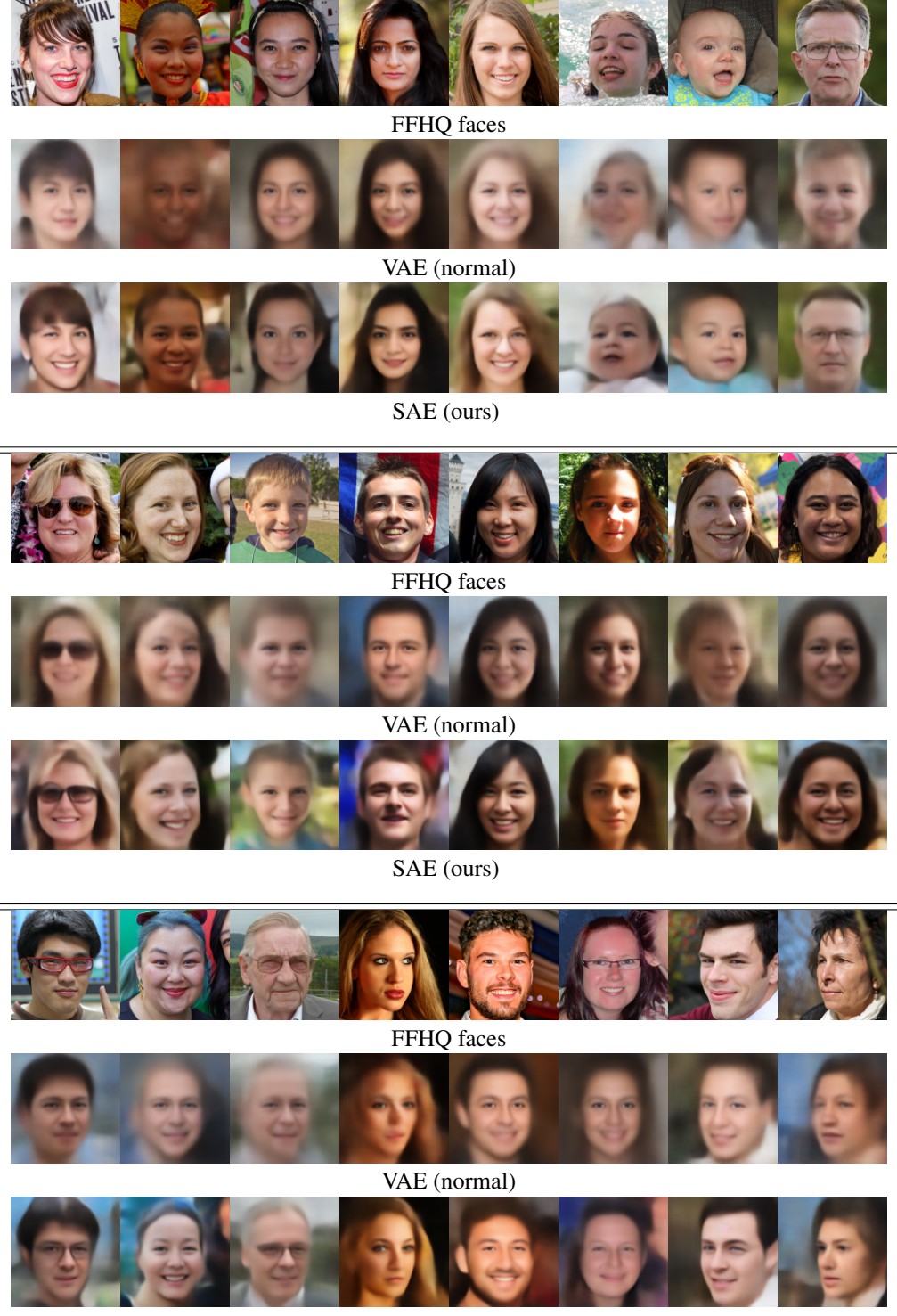

Figure 6: Reconstructed faces by VAE and SAE. Our SAE algorithm only uses the spherical constraint in equation (10) instead of priors.

## A.2    Sampling VAE and SAE on FFHQ

VAE (normal)                                               SAE (ours)

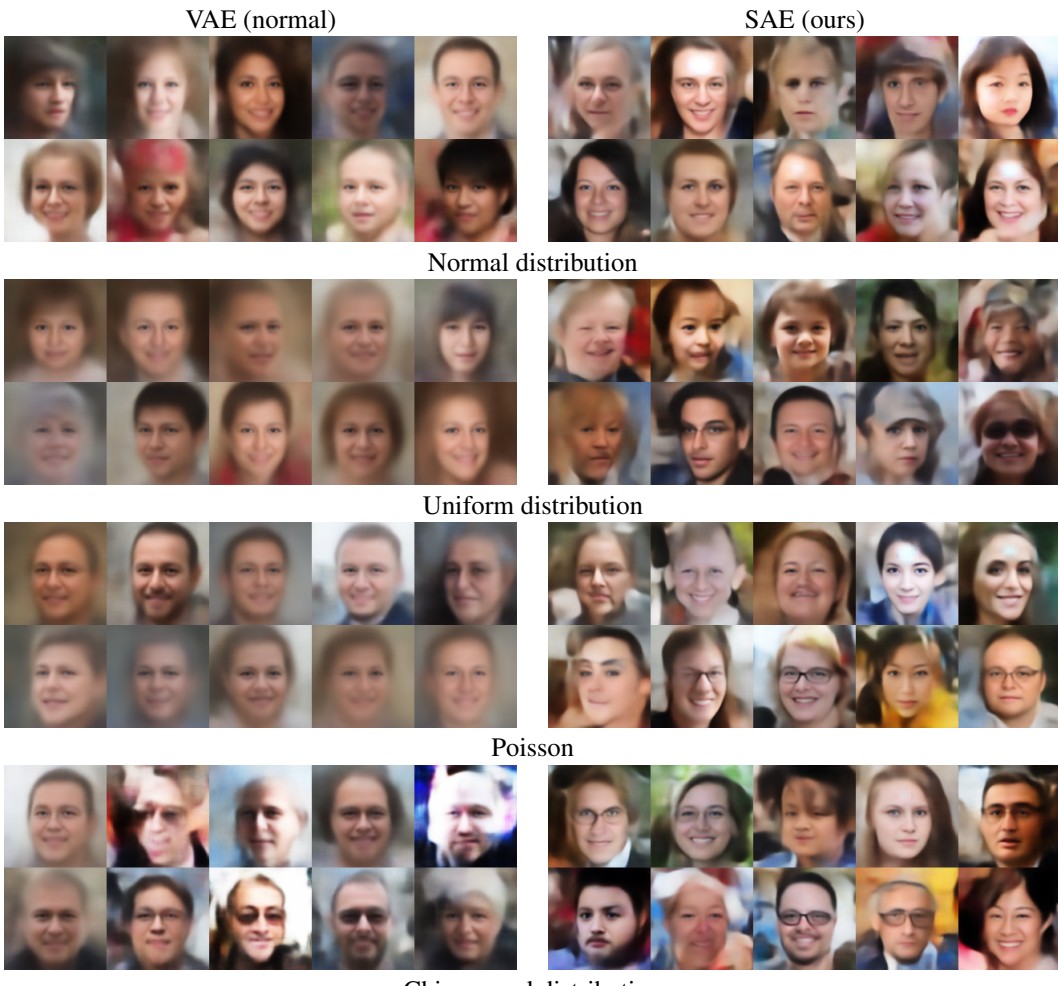

Normal distribution

Uniform distribution

Poisson

Chi-squared distribution

Figure 7: Generated faces with inputs of different priors. With the pre-trained decoders of VAE and SAE, the faces are generated with random vectors sampled from the four probability priors.

VAE (normal)               SAE (ours)

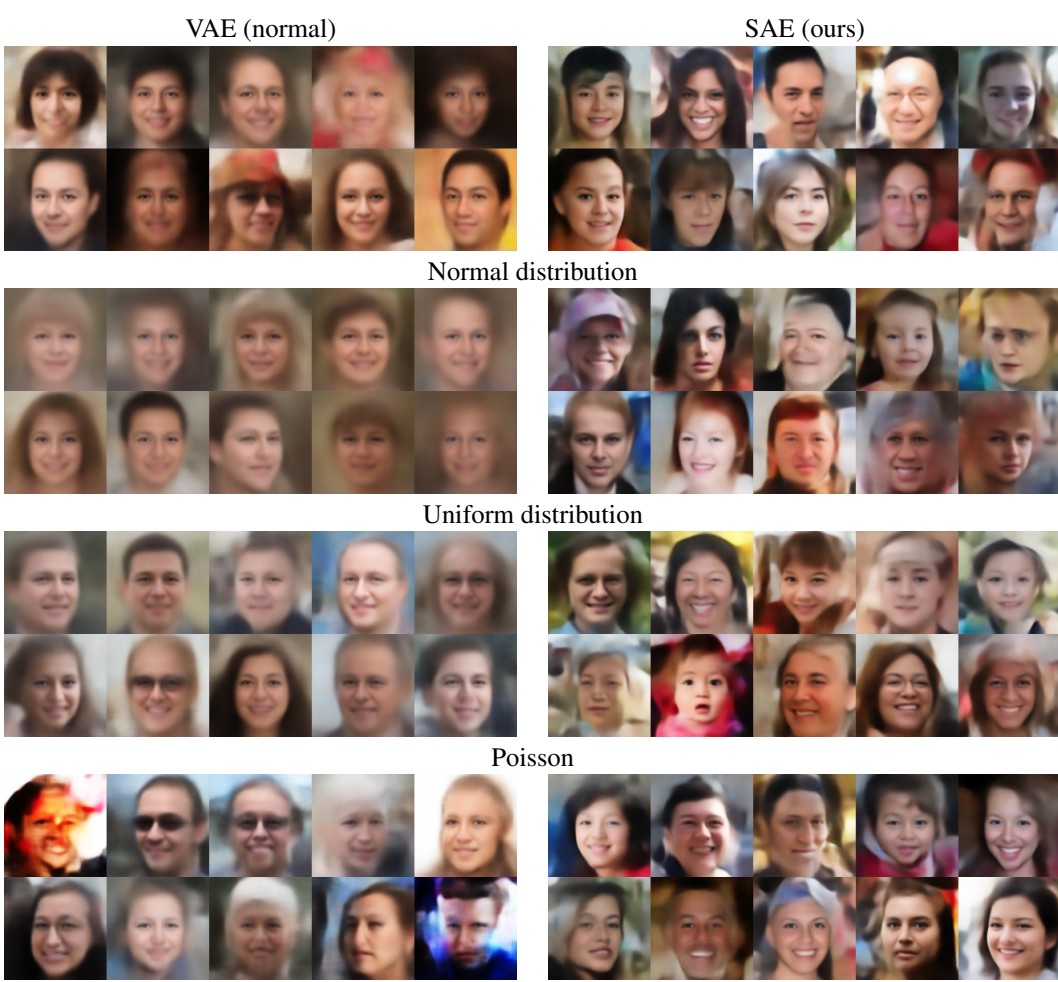

Normal distribution

Uniform distribution

Poisson

Chi-squared distribution

Figure 8: Generated faces with inputs of different priors. With the pre-trained decoders, the faces are generated with random vectors sampled from the four probability priors.

## A.3 RECONSTRUCTION ON CELEBA

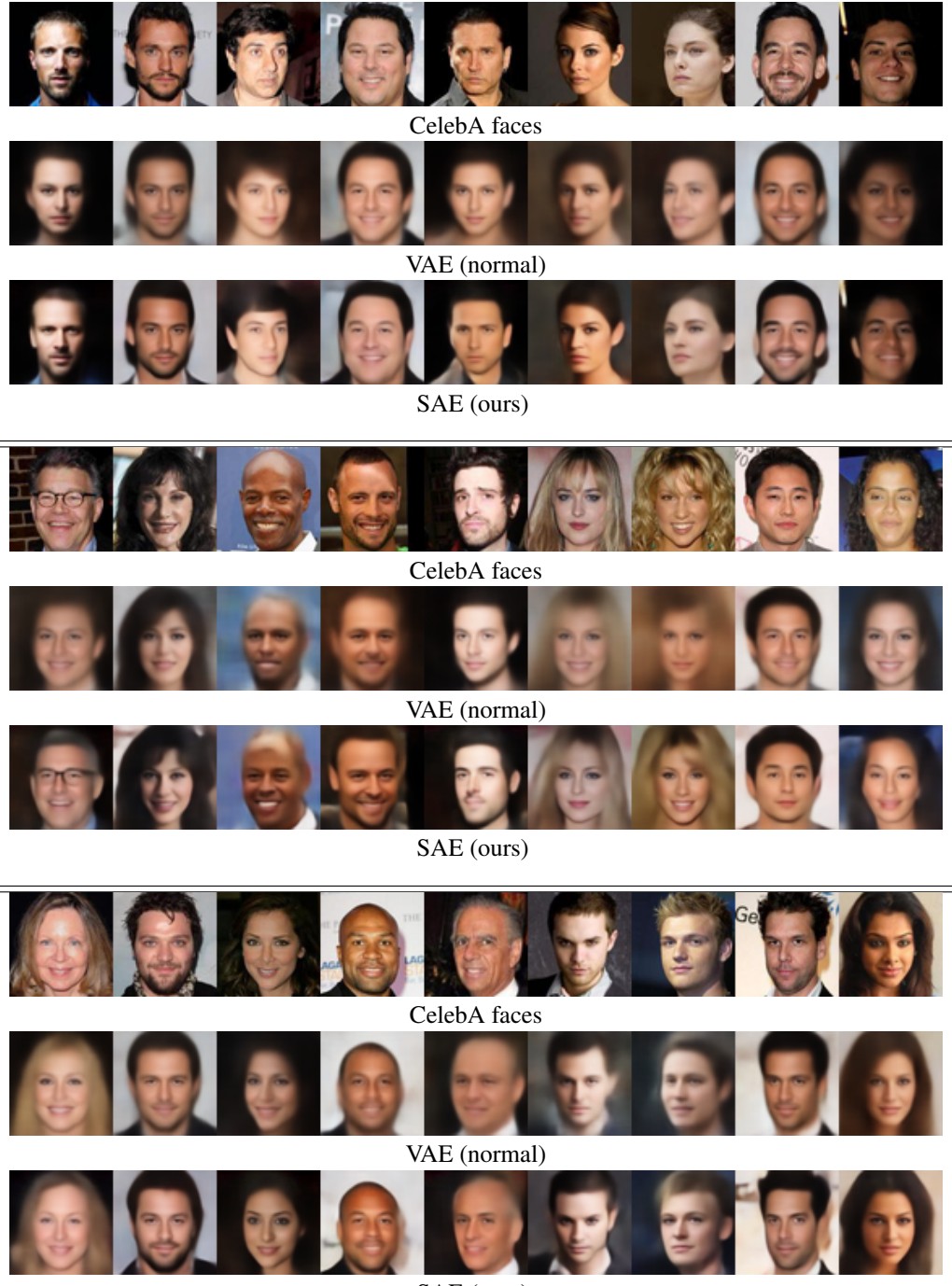

Figure 9: Reconstructed faces by VAE and SAE. Our SAE algorithm only uses the spherical constraint in (10) instead of priors.

## A.4 SAMPLING ON CELEBA

VAE (normal)                                    SAE (ours)

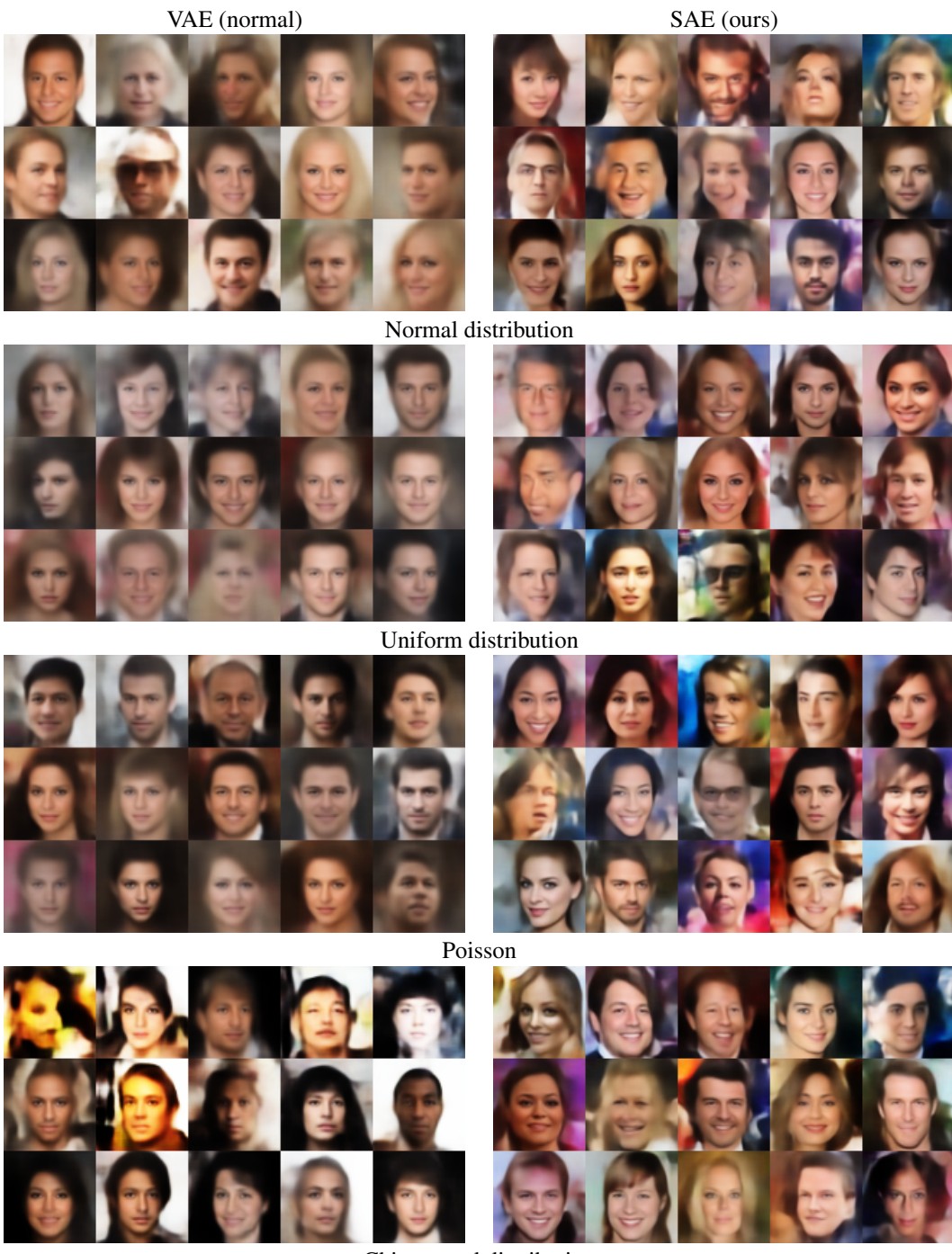

Normal distribution

Uniform distribution

Poisson

Chi-squared distribution

Figure 10: Generated faces with inputs of different priors. With the pre-trained decoders, the faces are generated with random vectors sampled from the four probability priors.

## A.5   RECONSTRUCTION ON MNIST

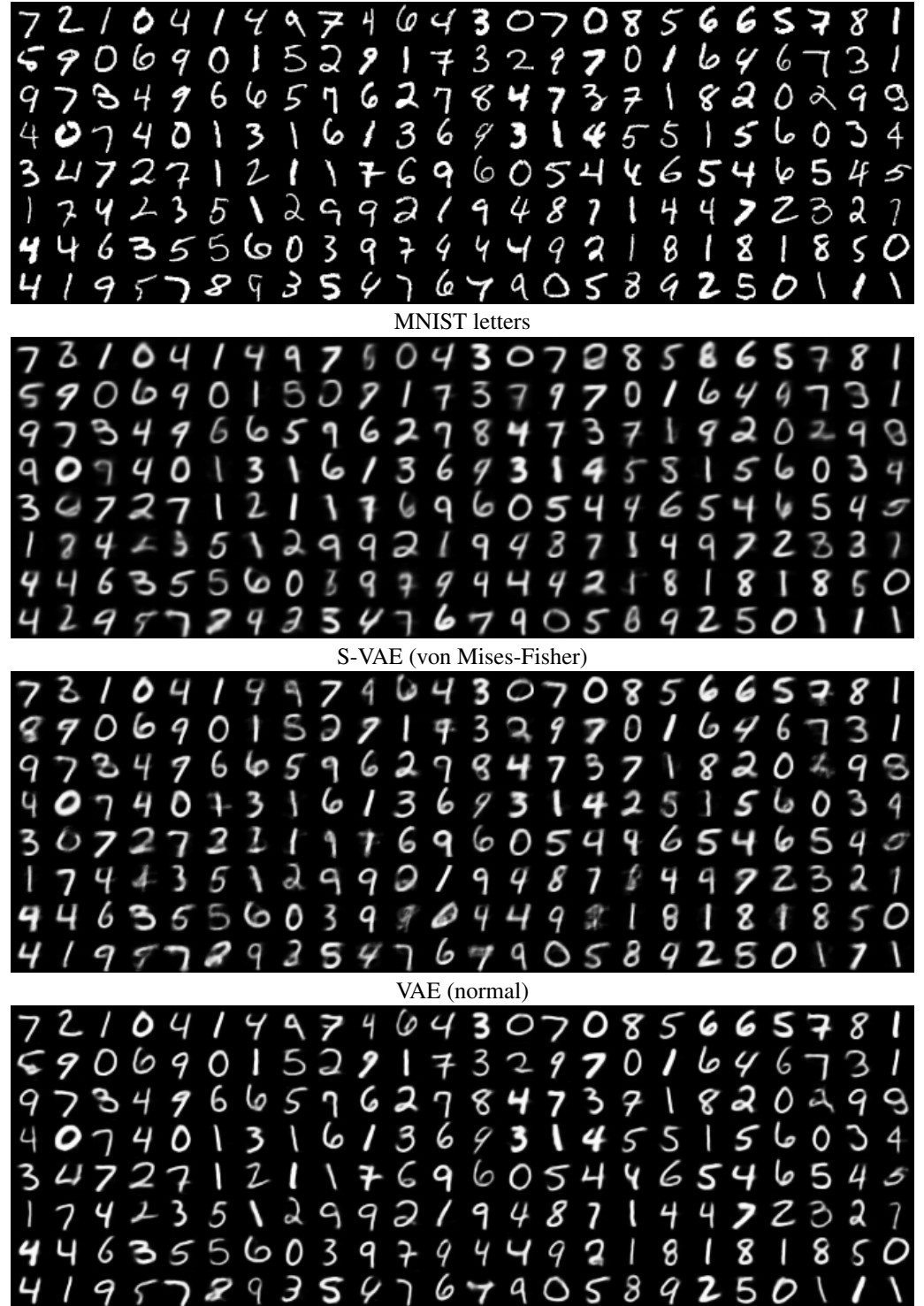

Figure 11: Reconstructed letters by VAEs with different priors on latent spaces. Our SAE algorithm only uses the spherical constraint in equation (10) instead of priors. For all experiments on MNIST, we take $d_z = 10$.

## A.6    SAMPLING S-VAE, VAE, AND SAE ON MNIST

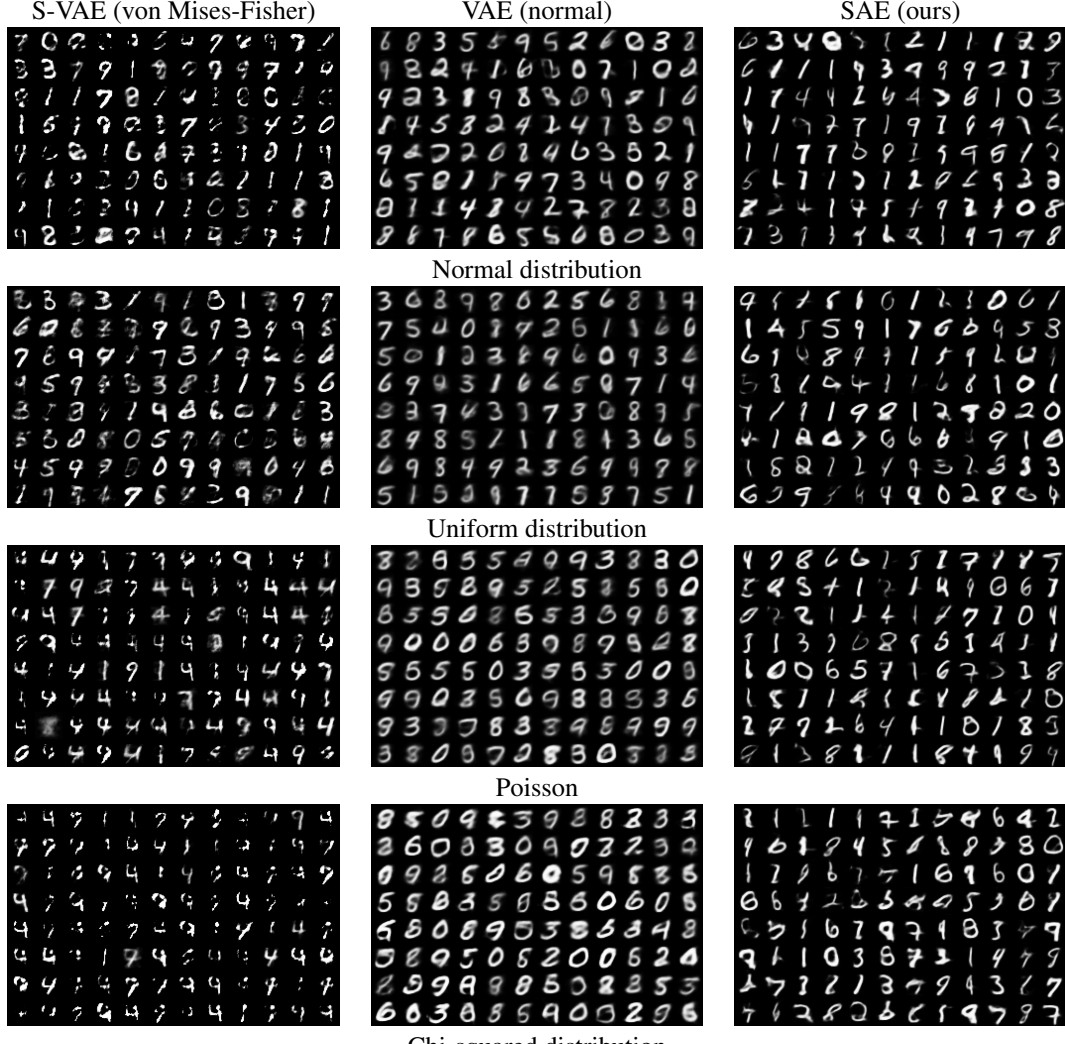

Figure 12: Generated letters with inputs of different priors. With the pre-trained decoders, the letters are generated with random vectors sampled from the four probability priors.

## A.7 VISUALIZATION OF INFERENCE

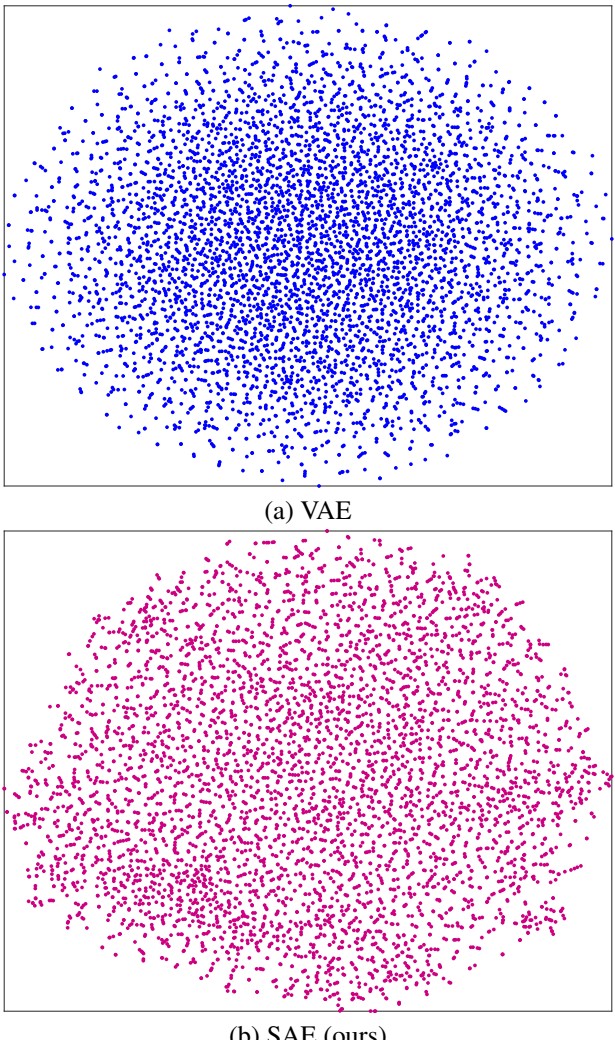

(a) VAE

(b) SAE (ours)

Figure 13: Visualization of inferred codes $z$ on CelebA with t-SNE. We randomly sample 5,000 faces from CelebA for illustration. The distribution of the latent codes from the variational inference (VAE) shows a standard normal one. However, the distribution of the latent codes from the spherical inference (SAE) is prone to be globally uniform while maintaining the variation of density.

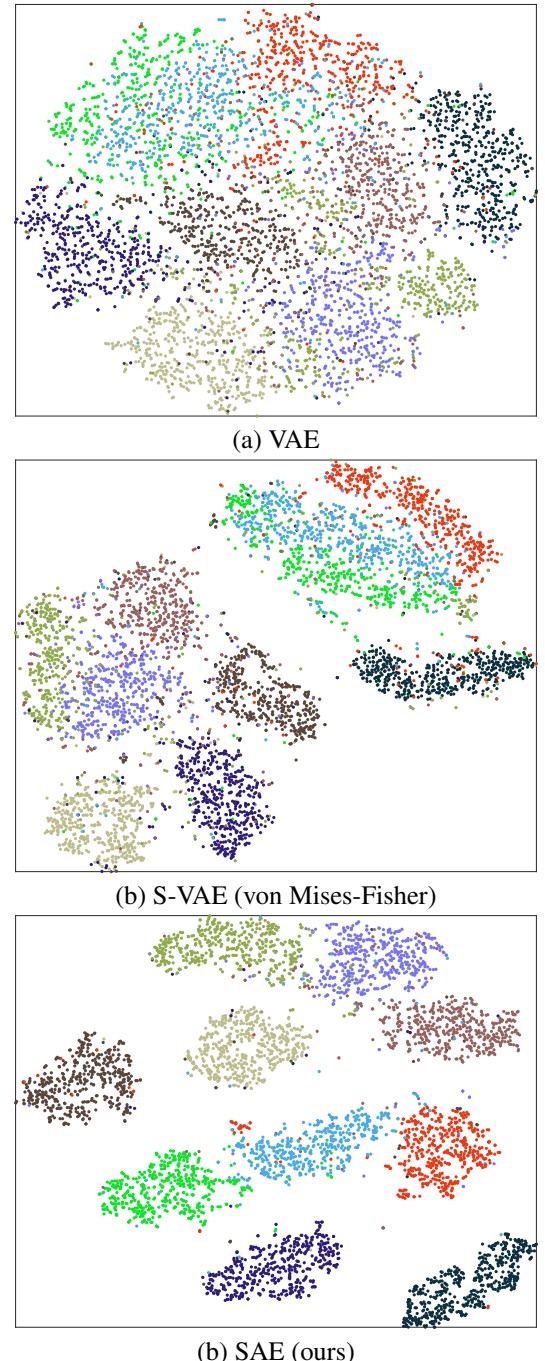

(a) VAE

(b) S-VAE (von Mises-Fisher)

(b) SAE (ours)

Figure 14: Visualization of inferred codes $z$ on MNIST with t-SNE. We randomly sample 500 letters from each class in MNIST to form the whole set for illustration. The latent codes derived from SAE is much better than that from VAE and S-VAE. The margins between different classes are wider, meaning that the latent codes from the spherical inference conveys more discriminative information in the way of unsupervised learning. This experiment also indicates that SAE captures the intrinsic structure of multi-class data better than VAE and S-VAE.

