# OpenReview forum: "Latent Variables on Spheres for Sampling and Inference"
_ICLR.cc/2020/Conference — Reject_

### Official Review · AnonReviewer1 · 2019-10-22
**Official Blind Review #1**

**Rating:** 6

**Review:**

This paper proposed an interesting idea that by regularizing the structure of the latent space into a sphere, we can free VAE from variantional inference framework.

However, here are several concerns about this paper:
1. is this a variant of the Wasserstein auto-encoder?
2. the image quality of VAE (CelebA) is not that bad in other VAE papers, maybe tuning the \beta-VAE can also achieve the same quantitative and qualitative results.
3. Can you visualize the latent space (z) for the CelebA dataset, also comparing with the results from VAE?

**Experience Assessment:**

I have published one or two papers in this area.

**Review Assessment: Checking Correctness Of Derivations And Theory:**

I assessed the sensibility of the derivations and theory.

**Review Assessment: Checking Correctness Of Experiments:**

I assessed the sensibility of the experiments.

**Review Assessment: Thoroughness In Paper Reading:**

I read the paper at least twice and used my best judgement in assessing the paper.

---

> ### Author Response · Authors · 2019-11-09
> **To Reviewer #1**
>
>
> Q1: “is this a variant of the Wasserstein auto-encoder?”
> A1: Our SAE algorithm is not a variant of WAE proposed in the following paper.
>
> Wasserstein Auto-Encoders
> https://arxiv.org/abs/1711.01558
>
> WAE minimized Wasserstein distance between the model distribution and the prior distribution. The algorithm reduces to adversarial learning via a discriminator in the latent space, which is similar to the following paper
>
> Adversarial Autoencoders
> https://arxiv.org/abs/1511.05644
>
> Like VAE, both Wasserstein autoencoder and adversarial autoencoder need a prior distribution to match. However, there is no loss imposed on the latent space to optimize for SAE. SAE does not need priors either. The object function is the reconstruction loss || x - \tilde{x} || with the spherical  constraint shown in equation (10). It is much simpler than Wasserstein autoencoder.
>
> In order to elucidate the unique property of random variables on spheres, we leveraged Wasserstein distance to derive Theorem 2. The Wasserstein distance here serves to establish a circumstance that the algorithm with the spherical constraint can be distribution-agnostic. We did not use Wasserstein distance for computation in SAE.
>
> Both Wasserstein autoencoder and adversarial autoencoder are very interesting and inspiring algorithms. We like these two works very much.
>
>
> Q2: “the image quality of VAE (CelebA) is not that bad in other VAE papers, maybe tuning the \beta-VAE can also achieve the same quantitative and qualitative results.”
> Q3: “Can you visualize the latent space (z) for the CelebA dataset, also comparing with the results from VAE?”
> A2: For data factors, the image quality of VAE depends on the image size and the image diversity. For face images, the large image size and more backgrounds in the image will make the data difficult to fit. We used the more challenging data of FFHQ available at  https://github.com/NVlabs/stylegan and the image size we used is 128x128.
>
> A3: We are conducting the experiment on CelebA of size 64x64 according to your advice. We will update the results when this complementary experiment is completed.

---

### Official Review · AnonReviewer3 · 2019-10-23
**Official Blind Review #3**

**Rating:** 1

**Review:**

Summary

This paper considers the L2 normalization of samples “z” from a given prior p(z) in Generative Adversarial Netowks (GAN) and autoencoders. The L2 normalization corresponds to projecting samples onto the surface of a unit-hypersphere. Hence, to attempt to justify this normalization, the authors rely on some already established results regarding high dimensional hyperspheres. In particular, the focus is on the fact that, the Euclidean distance between any given point on a hypersphere and another randomly sampled point on the hypersphere tends to a constant, when the number of dimensions goes to infinity. This result is then used to show that the Wasserstein distance between two arbitrary distributions on a hypersphere converges to a constant when the number of dimensions grows. Based on this result, the authors claim that projecting the latent samples onto the surface of a hypersphere would make GAN less sensitive to the choice of the prior distribution. Moreover, they claim that such normalization would also benefits inference, and that it addresses the issue of variational inference in VAE.

Main comments.

This paper is hard to follow and requires substantial improvements in terms of writing, owing to several grammatical and semantic issues. Moreover, there is a lack rigor; some important claims are supported neither by experiments nor by theoretical analysis. Experiments in the main paper are also weak. I can therefore not recommend acceptance. My detailed comments are below.

- An important claim in this paper is that the proposed approach “alleviates variational inference in VAE”. However, this requires clarification as well as theoretical/empirical justifications.
- In the introduction, it is stated that generated samples from VAE may deviate from real data samples, because “the posterior q(z|x) cannot match  the prior p(z) perfectly”. However, in VAE we do not expect the posterior to match the prior perfectly, as this would result in useless data representations or inference. Generation issues in VAE may rather be explained by the fact that, in this context we optimize a lower bound on the KL-divergence between the empirical data distribution and the model distribution. The latter objective does not penalize the model distribution if it puts some of its mass in regions where the empirical data distribution is very low or even zero.
- Theorem 2 (on the convergence of the Wasserstein distance (W2) on high dimensional hyperspheres) does not seem to hold if, for instance, P and P’ are empirical distributions with overlapping supports. Further, even when the above Theorem holds, the W2 distance may be relatively high since it is proportional to the square root of the number of samples.
- Moreover, why and how would Theorem 2 justify improved inference when projecting latent samples onto a hypersphere?
- Please consider revising the following statement in the introduction: “The encoder f in VAE approximates the posterior q(z|x)”. The encoder “f” in VAE parametrizes the variational posterior.
- Some typos,
	- Abstract, “… by sampling and inference tasks”  -- “on sampling …”
	- Introduction second paragraph after eq 2. “… it also causes the new problems” – “ … causes new problems”
	- Section 2.1, “For convenient analysis …” – “For a convenient …”
	- Second paragraph after Theorem 1. “… perform probabilistic optimizations … ” – “… optimization …”
	- Section 5.2, second paragraph. Is it Figure 9?

The main recommendations I would make are as follows.
- Consider revising the paper to improve its writing.
- Provide rigorous theoretical analysis and discussions to support the main claims.
- Improve experiments by including more datasets and baselines (e.g., hyperspherical VAE [1]), as well conduct more targeted experiments to give more insights regarding the effect of the L2 normalization on inference and generation.

[1] Davidson, Tim R., et al. "Hyperspherical variational auto-encoders." UAI, 2018.


**Experience Assessment:**

I have published in this field for several years.

**Review Assessment: Checking Correctness Of Derivations And Theory:**

I assessed the sensibility of the derivations and theory.

**Review Assessment: Checking Correctness Of Experiments:**

I assessed the sensibility of the experiments.

**Review Assessment: Thoroughness In Paper Reading:**

I read the paper thoroughly.

---

> ### Author Response · Authors · 2019-11-09
> **To Reviewer #3**
>
>
> Q1: “An important claim in this paper is that the proposed approach “alleviates variational inference in VAE”. However, this requires clarification as well as theoretical/empirical justifications”
> Q2: “Moreover, why and how would Theorem 2 justify improved inference when projecting latent samples onto a hypersphere?”
> A1 and A2: These two questions and related comments might be due to our inappropriate use of “alleviate” and the extensive meaning of inference beyond probability. Actually, there is no posterior inference and any priors involved in our SAE algorithm. It is the vanilla autoencoder subject to the spherical constraint shown in equation (10). So we said “thus freeing VAE from the approximate optimization of posterior probability via variational inference” and “Our algorithm is geometric and free from posterior probability optimization”. Indeed, “alleviates variational inference in VAE” is an inappropriate use in this scenario. We will correct this in the revised version.
>
> Besides, we use “inference” to refer to inferring (obtaining) z from the encoder, not only for “variational” inference or “probabilistic” inference. This might cause misunderstanding with habitual thinking in this field. This misunderstanding might be avoided by using “geometric inference”. We will note this meaning clearly in the revised version.
>
>
> Q3: “However, in VAE we do not expect the posterior to match the prior perfectly, as this would result in useless data representations or inference.”
> A3: We  understand your viewpoint about the model distribution and the prior distribution . “match the prior perfectly” does not mean the point-to-point correspondence. We refer to fitting distributions. The word “match” is also used in Wasserstein autoencoder (https://arxiv.org/abs/1711.01558), which is the same scenario to ours.
>
> Q4: “Theorem 2 (on the convergence of the Wasserstein distance (W2) on high dimensional hyperspheres) does not seem to hold if, for instance, P and P’ are empirical distributions with overlapping supports.”
> Q5: “Further, even when the above Theorem holds, the W2 distance may be relatively high since it is proportional to the square root of the number of samples.”
> A4: To make our theory much easier to understand, we directly gave the computational definition of Wasserstein distance in (8) and (9) rather than its original integral form. Thus, Theorem 2 is the direct result by substituting the conclusion of Lemma 1 into (8). It is very easy. About the correctness of Lemma 1, please refer to the elegant proof at http://faculty.madisoncollege.edu/alehnen/sphere/hypers.htm.
>
> Most theorems only hold under some conditions. Both Lemma 1 and Theorem 2 need a basic condition. The condition is that the points are drawn from spheres at RANDOM. To satisfy the condition, we use the operation of centerization in our SAE algorithm, which is motivated from central limit theorem in probability.
>
> In fact, it is straightforward to design the case to deny Lemma 1 and Theorem 2 if we bypass the condition. For instance, let Z1 be the set sampled from the spherical part in the open positive orthant and Z2 sampled from the spherical part in the open negative orthant. The third set Z3 is derived from Z2 by the small perturbation. Both Lemma 1 and Theorem 2 do not hold for the dataset { Z1, Z2, Z3}. But such samping violates the randomness needed. For SAE, the centerization is used to prevent such cases.
>
> A5: “the W2 distance may be relatively high since it is proportional to the square root of the number of samples.” is correct. However, it is logically wrong to use it to deny our theory, because all the W2 distances between two arbitrary random datasets still converge to be the same constant in Theorem 2 when the number of samples increases. The conclusion still holds in our paper.
>
> Q6: “Improve experiments by including more datasets and baselines (e.g., hyperspherical VAE [1]), as well conduct more targeted experiments to give more insights regarding the effect of the L2 normalization on inference and generation. ”
> A6: We failed to get the convergent results of hyper-Spherical VAE (S-VAE) on FFHQ faces of size 128x128. So we did not compare it in the current version. We are now running it on MNIST. The results will be updated in the revised version within several days.

---

> > ### Comment · AnonReviewer3 · 2019-11-15
> > **Response to the rebuttal**
> >
> > Thank you for your response.
> >
> > I went through the rebuttal and the revised version of the paper, and most of my original concerns remain unaddressed:
> >
> > - The positioning of the paper with respect to VAE, variational inference is confusing and even misleading.
> >
> > - Attributing generation issues in VAE to the fact that “the posterior q(z|x) is incapable of matching the prior distribution p(z) well” is not correct. In VAE we are not expecting q(z|x) to match p(z) well as this would result in useless inference (q(z|x) ignoring x).  Please refer to my original comments for more details. In their answer A3, the authors mentioned about Wasserstein autoencoder (WAE). I would like to emphasize that in WAE the objective is to match the “aggregated” posterior q(z) with the prior p(z), where  q(z) = \int q(x)q(z|x)dx, with q(x) denoting the empirical data distribution.  Indeed, we want q(z), but not q(z|x), to perfectly match p(z).
> >
> > - Assumptions under which Theorem 2 holds should be stated clearly. The sampling at random assumption is not enough if z, z’ are from empirical distributions on the sphere with overlapping supports.
> >
> > - While the result of Theorem 2 may justify “distribution robust-sampling”, it is not clear how and why would this Theorem justify improvement in inference when projecting z onto a hypersphere.
> >
> > - The revised version includes qualitative results for VAE and the proposed SAE on a new dataset (CelebA), as well as for hyper-Spherical VAE on MNIST. However, quantitative experiments remain weak.
> >
> > - The writing has been revised making it better than the initial version, yet the paper is still hard to follow, and further improvements are necessary.

---

### Official Review · AnonReviewer2 · 2019-10-23
**Official Blind Review #2**

**Rating:** 6

**Review:**

This paper proposes a novel autoencoder algorithm, named Spherical AutoEncoder (SAE). In this paper, the authors argue that the sphere structure has good properties in high-dimensional. To leverage the properties, proposed algorithm centerizes latent variables and projects them onto unit sphere. To show the empirical performance of the proposed approach, the authors perform image reconstruction and generation using FFHQ dataset and MNIST dataset.

Comments:
I think the proposed approach, using spherical latent space, is interesting and make sense.

- As mentioned in section 3.2, the proposed algorithm is reduced to standard autoencoder since it is free from posterior inference. Then, to clarify the algorithm, it seems necessary to provide the formulation of objective functions.
- Is the objective still valid or reasonable even it is derived from the equation (10) without posterior inference?
- How does the objective change when centerization and spherization are applied to the GAN?
- Compared with using von Mises-Fisher distribution in the vanilla VAE, the advantage of the proposed method is not clear. To my understanding, the main difference seems to be whether using lower bound with posterior inference or deterministic framework without such approximation. However, there are no theoretical or empirical results to show the benefit of the proposed method. If theoretical or empirical results with reasonable intuition is provided, it will make the proposed algorithm more valuable.

Questions:
- Compare to ProGAN and StyleGAN, is the contribution of the paper to applying centerization to GAN and centerization and spherization to autoencoder?
- What dimension do you use as latent dimension in the experiments?
- Does the choice of prior distribution affect the experimental results? If so, is there any compatible reason with the intuition of SAE?

Typo:
Under equation (10) in page 5: \tilde{z} should be \hat{z}.


**Experience Assessment:**

I have read many papers in this area.

**Review Assessment: Checking Correctness Of Derivations And Theory:**

I assessed the sensibility of the derivations and theory.

**Review Assessment: Checking Correctness Of Experiments:**

I assessed the sensibility of the experiments.

**Review Assessment: Thoroughness In Paper Reading:**

I made a quick assessment of this paper.

---

> ### Author Response · Authors · 2019-11-09
> **To Reviewer #2**
>
>
> Q1: “Then, to clarify the algorithm, it seems necessary to provide the formulation of objective functions.”,  “Is the objective still valid or reasonable even it is derived from the equation (10) without posterior inference?”
> A1: The objective function will be provided in the revised version. It is the reconstruction loss || x - \tilde{x} || subject to the spherical constraint on z (equation (10)). There are no posterior inference and no KL-divergence involved in our algorithm. It is very simple.
>
> Q2: “How does the objective change when centerization and spherization are applied to the GAN?”
> A2:  There is no extra objective when applied to GANs. Only centerization and spherization are needed.
>
> Q3: “Compared with using von Mises-Fisher distribution in the vanilla VAE, the advantage of the proposed method is not clear. To my understanding, the main difference seems to be whether using lower bound with posterior inference or deterministic framework without such approximation. However, there are no theoretical or empirical results to show the benefit of the proposed method.”
> A3: This might be the misunderstanding caused by that we didn’t explicitly write the objective function in the paper. We explain this in Q1. Our SAE algorithm is essentially different from S-VAE (hyper-Spherical VAE). The S-VAE is established on the principle of VAE. So, S-VAE has the drawbacks posed by VAE such as the approximation of posterior inference, the prior dependence, and the reparameterization trick for random variables. But SAE is distribution-agnostic with respect to Wasserstein distance, which is rigorously guaranteed by Theorem 2.
>
> Actually, we failed to get the convergent results of S-VAE on FFHQ faces of size 128x128. We are now running it on MNIST. The results will be updated in the revised version within several days.
>
>
> Q4: “Compare to ProGAN and StyleGAN, is the contribution of the paper to applying centerization to GAN and centerization and spherization to autoencoder?”
> A4: Both GAN and autoencoder need to use centerization and spherization on random variables. For ProGAN and StyleGAN, the authors empirically applied spherization on z in their code, which motivated our work. We also made it clear in the context of equation (3).
>
> Q4: “What dimension do you use as latent dimension in the experiments?”
> A4: We followed the experimental setting of StyleGAN. The 512-dimensional latent vectors are used for StyleGAN, VAE, and SAE on the face datasets including FFHQ and CelebA. For MNIST, we take the 10-dimensional latent codes.
>
> Q5: “Does the choice of prior distribution affect the experimental results? If so, is there any compatible reason with the intuition of SAE?”
> A5: This question might be another misunderstanding caused by Q1. Actually, there are no any priors involved in SAE during training. We used different priors to test the robustness of SAE and VAE after training was completed. We will make it clear in the revised version.

---

### Public Comment · ~Alex_Matthew_Lamb1 · 2019-10-16
**Some questions related to theory for spherical autoencoder**

I think this is an interesting paper and I was quite impressed by the results and elegance of the approach.  I have some thoughts about it from a conceptual point of view though.

1.  I'm curious if the SAE loss going to zero guarantees good samples in a theoretical sense.  I'm not sure if this is the case because during training the z's are always projected onto the sphere, but there is no requirement that they cover all of the points on the sphere.  So I could imagine a way of packing all of the z's seen during training into a small region on part of the sphere, having these points decode well, and then having all of the other regions decode to bad points.

Perhaps the inductive bias of the neural network makes this type of issue unlikely - in either case it makes it interesting that it seems to work so well.  (if it's the case it reminds me a bit of the cyclegan, where there is technically a way for the model to do something bad, but it doesn't happen as a result of the inductive bias from the architecture).

I think I have a particular construction that you might find interesting.  Let’s say that each real data point is binary, for example x in N^784 (as with binary MNIST).  I can encode this digit in a single number xb by laying out the digits: 0.0011010…1 (with each decimal point corresponding to that pixel position).

Now let c = sqrt((1 - 2*xb^2) / 2)

Then let’s say z = [xb, -xb, c, -c].

Regardless of xb, so long as it is between (-sqrt(0.5), sqrt(0.5)), this z will be centered and on the sphere.  I realize that this is extremely unlikely to be learnable by a NN, especially due to smoothness and its inductive biases.  However I still think it's at least interesting to think about.

2.  I think one reason SAE might work so well in practice is due to the asymmetry in the KL-divergence in the VAE objective, where you have KL(q(z|x) || p(z)).  It becomes unbounded and large if q(z|x) ever has support but p(z) doesn't have support.  By pushing q(z | x) onto the sphere, and because samples from p(z) are essentially always on the sphere, you guarantee that the KL is at least bounded.  In practice this might be enough to make KL(q(z|x)||p(z)) sufficiently small, especially because the SAE doesn’t have any incentive to concentrate the encoded points in particular parts of z-space, even though it hypothetically could.

---

> ### Author Response · Authors · 2019-10-18
> **about covering and inductive bias**
>
> Thanks for your very insightful comments, Alex.
>
> 1) About covering
>
> The case you raised is really challenging. We choose the vector centerization to afford randomness on the sphere. But how well this operation enforces z_i to cover the spherical surface as uniformly as possible is an important topic to study for spherical autoencoder (SAE).
>
> The vector centerization presented in our paper does have flaw. For example, the points on the spherical surface falling into the open positive orthant (z_i > 0) cannot be sampled. Considering that there are all 2^512 orthants in R^512, however, the open positive orthant only takes 1/2^512 part of the whole sphere. So, the negative effect is nearly trivial.
>
> We also figured out another way of sampling on the sphere to circumvent this problem. For any {z_1,...,z_i,...,z_n} drawn from arbitrary distributions, we can first project them on the sphere by z_i <-- z_i/norm(z_i). The projected points probably lie on some specific regions on the sphere.  Then we can randomly rotate these points on the sphere by a series of orthogonal matrices that are obtained by orthogonalizing random matrices via Gram–Schmidt process. In this way, we can get {z_1,...,z_i,...,z_n} that distributes randomly on the sphere as long as the rotation manipulations are sufficient.
> However, this method is not friendly to end-to-end learning for autoencoder. We do not use it in this paper.
>
> The vector centerization and spherization is the simplest way we get to realize our idea, even though it is not perfect. What is most important is that it is very easy to use in the end-to-end architecture of autoencoder.
>
> 2) About inductive bias
>
> Theorem 2 tells that SAE is distribution-agnostic with respect to Wasserstein distance. In other words, it has distributional inductive bias. However, it is very inspiring about your conjecture "Perhaps the inductive bias of the neural network makes this type of issue unlikely".
>
> Actually, your conjecture leads to the connection between random variables on the sphere and universal approximation theorem (UAT).  We also think that this is an alternative way of further exposing the deep reason why the simple spherical constraint can outperform the traditional variational inference. You may refer to the following paper, if interested.
>
> Spherical approximate identity neural networks are universal approximators
>  Zarita Zainuddin, Saeed Panahian Fard
> ICNC, 2014
>
> Your thoughts are quite inspiring. We will consider the topics you raised seriously for our future work.

---

### Author Response · Authors · 2019-11-14
**About the revised version**

The revised version has been updated. We revised the submission from the following eight aspects according to Reviewers' advice.

1. We explicitly wrote the objective function of SAE in equation (11). The reconstruction loss and the spherical constraint in equations (10) and (11) are all operations in SAE. There are no variational inference, no probabilistic optimization, and no priors involved in SAE during training.

2. To make the meaning of the word "inference" clear, we named the inference in SAE as the spherical inference. As opposed to the variational inference, the spherical inference  is deterministic during training. But the decoder of SAE is rather robust to various priors for sampling after training. We made this expression clear in the paper to avoid misunderstanding with the variational inference.

3. We added the discussion for Wasserstein autoencoder, adversarial autoencoder, and beta-VAE in the related work.

4. We also compared VAE and SAE on CelebA.  We need to note that the quality of generated faces by VAE and SAE will be both improved if we use the face images of only cropping facial parts for the experiments. But such data are not sufficient to test the robustness of the algorithms against the variations of the entire facial features.

5. We compared hyper-Spherical VAE (S-VAE) with SAE on MNIST using the official code at https://github.com/nicola-decao/s-vae-tf.

6. We provided the visualization results of latent codes from VAE, S-VAE, and SAE on CelebA and MNIST. This visualization clearly shows the superiority of the spherical inference in SAE.

7. We re-arranged images in Figure 5 to save more space for the new contents. The experimental results in this Figure are kept the same as the previous version.

8. We corrected the typos, polished the writing,  and made the paper more readable.

All the complementary experimental results were attached in Appendix.

---

### Public Comment · ~Alokendu_Mazumder1 · 2024-01-17
**Inference Results**

Happy New Year and Hope you're well!

I have a question regarding the understanding of your model at the inference stage:

1. During reconstruction, the un-normalized latent vector is fed directly to the decoder?

Thanks and Best Regards,

Alokendu Mazumder
PhD Scholar
Indian Institute of Science, Bengaluru, India

---

### Decision · Program_Chairs · 2019-12-19

**Decision:**

Reject

**Comment:**

This paper proposes to improve VAE/GAN by performing variational inference with a constraint that the latent variables lie on a sphere. The reviewers find some technical issues with the paper (R3's comment regarding theorem 3). They also found that the method is not motivated well, and the paper is not convincing. Based on this feedback, I recommend to reject the paper.